# Modeling Catalyst-Free Growth of III-V Nanowires: Empirical and Rigorous Approaches

**DOI:** 10.3390/nano13071253

**Published:** 2023-04-01

**Authors:** Vladimir G. Dubrovskii

**Affiliations:** Faculty of Physics, St. Petersburg State University, Universitetskaya Emb. 13B, 199034 St. Petersburg, Russia; dubrovskii.ioffe@mail.ru

**Keywords:** III-V nanowires, selective area growth, radial growth, adatom diffusion, nanowire length and radius, modeling

## Abstract

Catalyst-free growth of III-V and III-nitride nanowires (NWs) by the self-induced nucleation mechanism or selective area growth (SAG) on different substrates, including Si, show great promise for monolithic integration of III-V optoelectronics with Si electronic platform. The morphological design of NW ensembles requires advanced growth modeling, which is much less developed for catalyst-free NWs compared to vapor–liquid–solid (VLS) NWs of the same materials. Herein, we present an empirical approach for modeling simultaneous axial and radial growths of untapered catalyst-free III-V NWs and compare it to the rigorous approach based on the stationary diffusion equations for different populations of group III adatoms. We study in detail the step flow occurring simultaneously on the NW sidewalls and top and derive the general laws governing the evolution of NW length and radius versus the growth parameters. The rigorous approach is reduced to the empirical equations in particular cases. A good correlation of the model with the data on the growth kinetics of SAG GaAs NWs and self-induced GaN NWs obtained by different epitaxy techniques is demonstrated. Overall, the developed theory provides a basis for the growth modeling of catalyst-free NWs and can be further extended to more complex NW morphologies.

## 1. Introduction

III-V and III-nitride NWs and NW heterostructures of different types are promising building blocks for fundamental research of semiconductor properties at the nanoscale and applications in nanoelectronics and nanophotonics [1,2,3,4,5,6]. Due to a small footprint of NWs in contact with a dissimilar substrate, elastic strain induced by lattice mismatch can be relaxed without forming misfit dislocations [3,4,5]. This property is very attractive for the realization of III-V NW device structures directly on Si substrates [6]. Bottom-up approaches for the synthesis of freestanding III-V NWs include Au-catalyzed VLS growth [7,8], self-catalyzed VLS growth in which Au is replaced with a group III metal [9], or catalyst-free growth [10,11,12,13,14,15,16,17,18,19,20,21,22,23,24,25,26,27,28,29]. The deposition techniques used for catalyst-free NW growth include molecular beam epitaxy (MBE) [10,11,12,13,14,15,20,21,22,23,29] and vapor phase epitaxy techniques (VPE) such as metal-organic vapor phase epitaxy (MOVPE) [18,24,25,26,27,28,30,31,32] and hydride vapor phase epitaxy involving chloride group III precursors (HVPE) [16,17]. GaN NWs can be grown by MBE on unpatterned Si_x_N_y_ layers covering Si(111) substrates by the self-induced mechanism [10,11,12,13,14,15], whereby isotropic Volmer–Weber growth is transitioned to anisotropic NW growth at a critical island radius of about 5 nm [12,13]. This self-induced growth method allows one to avoid the unwanted contamination of GaN NW material with a growth catalyst in the VLS technique and costly lithographic patterning of a substrate in the SAG technique. Additionally, GaN NWs evolving from the Volmer–Weber islands are thinner and denser than VLS or SAG NWs. A similar self-induced growth mechanism is observed in the HVPE of InAs NWs on Si(111) substrates [16]. SAG of III-V and GaN NWs in patterned pinholes in SiO_x_ mask layers on different substrates [18,19,20,21,22,23,24,25,26,27,28,29,30,31,32] allows for the fabrication of regular arrays of NWs with pre-defined locations and pitch. In some cases, the true SAG and VLS growth can even coexist in one sample depending on the NW radius, pitch, and growth conditions [31].

Morphological control over the ensembles of catalyst-free NWs, including the evolution of their length, radius, volume, aspect ratio, and shape with time, requires kinetic modeling. Theoretical understanding of catalyst-free NW growth was partly based on far more developed models for group III limited VLS growth of III-V NWs [33,34,35,36,37]. In particular, the inverse length-radius correlations observed in the ensembles of self-induced GaN NWs [11] or regular arrays of SAG GaAs [25] and InAs [27] NWs were attributed to surface diffusion of group III adatoms from the NW sidewalls or mask to the top. The decrease of GaAs NW length in dense arrays [25] was explained by the pitch-dependent competition of NWs for group III flux. Many efforts were put into modeling the additional group III flux re-emitted from the mask surface, which is similar to VLS and catalyst-free NWs [36,37]. Narrowing of the length distribution of GaN NWs was studied in a more complex scenario involving a material exchange between the NWs [38]. A good understanding of the selectivity maps was achieved for III-V [20,39] and GaN [20] nanomaterials, including vertical and planar NWs in terms of temperatures and V/III flux ratios separating the SAG zones from the zones of no-growth or parasitic growth on a mask. Models of Refs. [40,41,42] were based on a phenomenological approach for treating the simultaneous axial and radial growths of catalyst-free III-V and GaN NWs and resulted in the scaling power-law dependencies of the NW length and radius on the growth time. The quantitative model for the axial growth rate of self-induced GaN NWs [43] was similar to the models for the diffusion-induced growth of VLS III-V NWs [33], where the catalyst droplet was replaced by the flat top facet. More rigorous studies based on the stationary diffusion equations for different populations of group III adatoms, such as described in Refs. [33,44] for VLS NWs, are solely lacking in the literature.

In this work, we try to fill the gap by presenting an empirical approach which unifies and extends the previous phenomenological models for catalyst-free growth [25,27,40,41,42] and developing a rigorous theory based on the diffusion equations for group III adatoms on the NW sidewall and top facet. Additionally, we take into account the secondary group III flux re-emitted from the substrate surface and NW sidewalls. Special attention is paid to some effects, which have not been studied before to our knowledge. First, in the absence of a catalyst droplet on top of self-induced or SAG NWs, the axial NW growth on the top facet proceeds via step flow, similar to the radial growth on the side facets. The two-step flows are synchronized due to material exchange by surface diffusion through the top edge, which depends on the NW length and radius. Therefore, surface diffusion and step flows on the side and top facets should be considered simultaneously versus the NW dimensions and growth parameters. Second, self-induced GaN NWs [11,12,13,14,15,41,42,43] and some SAG III-V NWs [16,30] are grown at high temperatures, with enhanced desorption of group III species. This leads to short diffusion lengths of group III adatoms on different surfaces. For example, the diffusion length of Ga adatoms on the sidewalls of GaN NWs was estimated at around 40 nm for temperatures around 780 °C [11,14,43]. The radius of catalyst-free III-V and III-nitride NWs may reach more than 200 nm [17,24,29,30], which may be much larger than the diffusion length on the top facet. In this case, most group III adatoms diffusing from the NW sidewalls to the top facet will evaporate rather than incorporate into the growing step. Some atoms impinging onto the top facet will also be lost for NW growth. Enhanced desorption of Ga atoms from thick enough GaN NWs has been reported, for example, in Refs. [17,43], but no theoretical models have been proposed to assess the effect in catalyst-free NWs. Third, the measured length-radius correlations of catalyst-free III-V and GaN NWs, which extended radially from the very beginning of growth [11,25,27,43], were explained using theoretical models originally developed for VLS III-V NWs having a time-independent radius [33,34]. Strictly speaking, such considerations are incorrect and maybe only used qualitatively or for catalyst-free NWs with almost negligible radial growth. These aspects of the catalyst-free growth process are taken into account in both empirical and rigorous approaches, which yield the same growth laws in the limiting cases. The model is used for fitting some experimental data on the axial and radial growths of GaAs [30] and GaN [42] NWs.

The paper is organized as follows. In Section 2, we present an empirical approach which establishes three distinct modes of the axial and radial NW growths for thin and short NWs, thin and long NWs, and thick and long NWs. The material fluxes of group III atoms are considered for individual NWs. The additional re-emitted flux of group III species is introduced later. Section 3 deals with the rigorous approach based on the coupled diffusion equations for the four populations of group III adatoms on each side of the steps propagating on the NW top and side facets. The symmetric boundary conditions for the adatoms at the NW top and bottom are shown to be essential for the specific growth laws discussed in Section 2. Taking an average of the diffusion fluxes, we obtain a refined growth picture, which is reduced to the three stages of Section 2 in the limiting cases. In Section 4, the length and radius evolution of NWs is considered in detail versus the growth parameters and template geometries. In Section 5, we consider some experimental data on the MOVPE SAG of GaAs NWs and MBE GaN NWs forming via the self-induced nucleation mechanism and fit the data using the developed model.

## 2. Empirical Approach

Catalyst-free NW growth is considered under the following assumptions. First, the NW growth under group V-rich conditions should be limited by the direct impingement, evaporation, and surface diffusion of group III species, as observed in MBE of self-induced GaN NWs [11,12,13,14,15,21,22,23], HVPE of InAs [16] and III-nitride [17] NWs, SAG of GaN NWs by MOVPE [24], SAG of InAs NWs by MBE [29], and SAG of different III-V NWs by MOVPE [25,26,27,28,30,32]. Group III’s limited diffusion-induced growth is also typical for Au-catalyzed VLS III-V NWs [33,34,35,36] unless the droplet gets very rich in group III atoms. In contrast, Ga-catalyzed VLS growth is limited by the adsorption–desorption kinetics of group V atoms [9,33]. Second, we consider untapered NWs with vertical sidewalls, such as self-induced GaN NWs of Refs. [11,12,13], self-induced InAs NWs of Ref. [16], SAG GaN NWs of Refs. [23,24], and most SAG III-V NWs grown by MOVPE [25,26,27,30,31,32]. Tapered [29] or inversely tapered [15] geometries of catalyst-free NWs, which are also often observed in VLS III-V NWs [35,45], should be studied separately. Untapered NW geometry is central for some considerations while using cylindrical geometry is not critical. For example, the coefficient π in all the expressions for cylindrical NWs should be changed to $3\sqrt{3}/2$ for hexahedral NWs. Third, both axial and radial NW growth rates are assumed limited by the material transport into the propagating steps rather than their nucleation. This assumption is standard in the transport-limited NW growth models [33,34,35,40,42]. Fourth, group III re-emission from the mask surface [36,37,45,46,47] is considered the main mechanism of material exchange between the NWs and the substrate. This assumption is standard for SAG of III-V NWs by MBE [36,37] and should be relevant for high-temperature MOVPE and HVPE growths [16,17,30,32]; it was also demonstrated for Ga-catalyzed GaP NWs grown by MBE on patterned SO_2_/Si(111) substrates at 600 °C [45]. For self-induced GaN NWs grown by MBE at high temperatures around 800 °C [11,12,13,14,15], the diffusion length of Ga on the NW sidewalls is so small that any contribution of surface diffusion from the substrate surface can be safely neglected after a short initial step [41,42,43]. This is further supported by very long incubation times preceding the growth of GaN islands and NWs (up to several hours) [13,48]. However, desorbed Ga atoms can subsequently contribute to NW growth. If group III adatoms are able to diffuse on the substrate surface, as in Refs. [35,49,50,51], one should consider the surface diffusion of adatoms rather than their re-emission, which will change the growth kinetics in the initial stage. Furthermore, the morphological evolution can be affected by negative diffusion flux from NW to the mask surface [49,50].

Let us first consider the catalyst-free growth of individual NW without material exchange with the substrate surface or neighboring NWs. High growth temperatures enhance the desorption of group III atoms from different surfaces, which was neglected in the modeling of MBE growth without any loss of group III atoms from NWs [45,46]. If v is the arrival rate of group III atoms onto the top facet, and $v_{f}$ is the arrival rate onto the side facets (in nm/s), the corresponding two-dimensional (2D) growth rates on these facets equal vΦ and $v_{f}\Phi_{f}$, with Φ≤1 and $\Phi_{f}\leq 1$ as the fractions of desorbed atoms. Then, the total flux onto the top facet equals $\pi R^{2}v\Phi$, and the total flux onto the sidewalls equals $2\pi RLv_{f}\Phi_{f}$, with L and R as the length and radius of cylindrical NW. In the empirical approach, the fluxes $c\pi R^{2}v\Phi$ and $a2\pi RLv_{f}\Phi_{f}$ are used for the axial growth, while the flux $b2\pi RLv_{f}\Phi_{f}$ is used for the radial growth, yielding
$$\frac{dL}{dt}=cv\Phi +a\frac{2L}{R}v_{f}\Phi_{f},$$
(1)$$\frac{dR}{dt}=bv_{f}\Phi_{f}.$$

The factors a, b, and c depend differently on L and R in different stages of growth, as illustrated in Figure 1.

In stage 1, “short” and “thin” NW collect adatoms from the whole top facet, corresponding to c=1. The collection lengths on the sidewalls of short NW should be proportional to its length L, meaning that a=const and b=const, as previously considered in Ref. [40]. The choice of a and b in this stage is not obvious and depends on the NW geometry and the number of steps progressing n their sidewalls. Untapered NW geometry can be maintained if only one step propagates on the NW sidewalls at a time [42]. Echelon of steps starting from the base, with a constant separation between them, would lead to positive tapering, with a tapering angle determined by this separation. Conically tapered shells growing around cylindrical cores are often observed in Au-catalyzed III-V NWs [52]. More complex pencil-like shapes are also possible [35]. Step bunching may lead to one step, which is thicker than monolayer, as modeled in Ref. [53] and observed experimentally in self-induced GaN NWs [14]. Here, we choose a and b specifically for untapered NWs with one monolayer step on the NW sidewalls at any time, as shown in Figure 1a–c. Assuming symmetrical profiles of adatom concentrations on each side of the steps [54], as shown in Figure 1a, we conclude that the top step collects the sidewall adatoms from an average length of L/4 at the NW top. Similarly, the side step collects, on average, half of all adatoms from the NW sidewalls from a length of L/2. The rest of the adatoms diffuse from the bottom part of the NW (with an average length of L/4) to the mask surface and subsequently evaporate. These considerations yield
(2)c=1, a=1/4, b=1/2,
and hence
$$\frac{dL}{dt}=v\Phi +\frac{L}{2R}v_{f}\Phi_{f},$$
(3)$$\frac{dR}{dt}=\frac{v_{f}\Phi_{f}}{2}.$$

In stage 2, “long” and “thin” NW continues collecting material from the whole top facet, but the sidewall collection areas are reduced. At $L\geq 4\lambda_{f}$, with $\lambda_{f}$ as the desorption-limited diffusion length of group III adatoms on the NW sidewalls, adatoms can reach the NW top only from the upper part of length $\lambda_{f}$. The side step collects adatoms from a length of $2\lambda_{f}$ on average, as shown in Figure 1b. For similar desorption-limited diffusion lengths on the NW sidewalls and top, this stage is relevant only for NWs with high enough aspect ratios L/R, acquired in stage 1. With
(4)$$c=1,\;a=\frac{\lambda_{f}}{L},\;b=\frac{2\lambda_{f}}{L},$$
the growth law in stage 2 is given by
$$\frac{dL}{dt}=v\Phi +\frac{2\lambda_{f}}{R}v_{f}\Phi_{f},$$
(5)$$\frac{dR}{dt}=\frac{2\lambda_{f}}{L}v_{f}\Phi_{f}.$$

Without the radial growth (which corresponds to $\lambda_{f}/L\to 0$ in our model), the axial growth rate is reduced to the simplest diffusion-induced law for long enough VLS [33,34,35] or catalyst-free [43] NWs at a constant radius.

In stage 3, the radius of “long” and “thick” NWs becomes much larger than the diffusion length of group III adatoms on the top facet λ. In this case, adatoms are collected only from a ring of width 2λ, as shown in Figure 1c. If 2D island starts in the center of the top facet, sidewall adatoms diffusing from the top cannot reach it before evaporation. The side facet collects material as in stage 2. These considerations yield
(6)$$c=\frac{2\lambda }{R},\;a=0,\;b=\frac{2\lambda_{f}}{L},$$
corresponding to the growth law
$$\frac{dL}{dt}=\frac{2\lambda }{R}v\Phi ,$$
(7)$$\frac{dR}{dt}=\frac{2\lambda_{f}}{L}v_{f}\Phi_{f}.$$

Similar equations were previously obtained in Ref. [42] from different considerations.

The additional flux of group III atoms re-emitted from the substrate surface and neighboring NWs cannot be calculated exactly. We will estimate it following the method of Refs. [46,47], generalized to systems with desorption. The total current F (in nm^3^/s) of group III atoms entering the NW by different kinetic pathways is used to increase the NW volume according to
(8)$$F=\frac{d}{dt}\left(\pi R^{2}L\right)=\pi R^{2}\frac{dL}{dt}+2\pi RL\frac{dR}{dt}=F_{dir}+F_{des}$$

Here, $F_{dir}$ is the direct vapor current, and $F_{des}$ is the secondary current which is initially re-emitted or desorbed from the substrate surface and NWs and subsequently lands on the NW surfaces. According to Equation (1), the direct current is given by
(9)$$F_{dir}=S_{nw}v\Phi ,\;S_{nw}=c\pi R^{2}+\left(a+b\right)2\pi RL\frac{v_{f}\Phi_{f}}{v\Phi },$$
where $S_{nw}$ stands for the effective NW surface area, which collects group III atoms. Both $F_{dir}$ and $S_{nw}$ account for desorption through the Φ and $\Phi_{f}$ factors. The secondary current $F_{des}$ is estimated using the material balance
(10)$$v=\frac{F_{dir}}{\xi P^{2}}+v_{des},\;F_{des}=S_{nw}^{\prime }v_{des}\Phi .$$

Here, $v_{des}$ is the desorption flux re-emitted from the mask and NW surfaces, and $\xi P^{2}$ is the unit cell area per NW, determined by the pitch P and geometrical factor ξ (for example, ξ=1 for square array of pinholes) [45,46,47]. For self-induced NWs [12,13,14,15,16], $\xi P^{2}$ is related to the number density of NWs N according to $N=1/\left(\xi P^{2}\right)$. The effective NW surface area exposed to re-emitted flux, $S_{nw}^{\prime }$, depends on the epitaxy technique and the re-emission model. In VPE techniques, $S_{nw}^{\prime }$ is the same as $S_{nw}$ in the first approximation [47]. In the directional MBE method, specular re-emission model of Ref. [45] results in zero secondary flux landing on the flat NW top and the same sidewall collection area as for the direct flux [46]. Using Equation (9) for $F_{dir}$ in Equation (10) and the above considerations for $S_{nw}^{\prime }$, we obtain
$$F_{des}=\left(1-\Phi \frac{S_{nw}}{\;\xi P^{2}}\right)S_{nw}^{\prime }v\Phi \;\mathrm{for}\;\Phi S_{nw}\leq \;\xi P^{2}\;\mathrm{and}\;0\;\mathrm{for}\;\Phi S_{nw}>\xi P^{2},$$
(11)$$S_{nw}^{\prime }=S_{nw}\;\mathrm{in}\;\mathrm{VPE}\;\mathrm{and}\;S_{nw}^{\prime }=\left(a+b\right)2\pi RL\frac{v_{f}\Phi_{f}}{v\Phi }\;\mathrm{in}\;\mathrm{MBE}.$$

The secondary current becomes zero when $\Phi S_{nw}$ reaches $\xi P^{2}$ because NWs cannot collect more material than is sent from vapor.

Summing up the direct and secondary currents, the total material current entering the NW is given by
(12)$$F=\left[S_{nw}+\left(1-\Phi \frac{S_{nw}}{\;\xi P^{2}}\right)S_{nw}^{\prime }\right]v\Phi ,\;\Phi S_{nw}\leq \;\xi P^{2},F=F_{max}=\xi P^{2}v,\;\Phi S_{nw}>\xi P^{2},$$
which is reduced to the results of Refs. [46,47] at Φ=1. Despite its simplicity, this expression correctly predicts that the total current saturates at its maximum value of $\xi P^{2}v$ when $\Phi S_{nw}$ reaches $\xi P^{2}$, meaning that the NW array collects all group III atoms arriving from vapor. Deconvolution of the total material current into the currents used for the axial and radial growths using Equations (8), (9) and (11) gives the modified growth law:(13)$$\frac{dL}{dt}=cv\Phi \left[1+\omega \left(1-\Phi \frac{S_{nw}}{\;\xi P^{2}}\right)\right]+a\frac{2L}{R}v_{f}\Phi_{f}\left(2-\Phi \frac{S_{nw}}{\;\xi P^{2}}\right),\omega =1\;\mathrm{in}\;\mathrm{VPE}\;\mathrm{and}\;0\;\mathrm{in}\;\mathrm{MBE},$$
(14)$$\frac{dR}{dt}=bv_{f}\Phi_{f}\left(2-\Phi \frac{S_{nw}}{\;\xi P^{2}}\right).$$

The factors a, b, and c in stages 1, 2, and 3 remain the same as before. These equations apply as long as $\Phi S_{nw}\leq \;\xi P^{2}$. In the asymptotic stage 4 proceeding at $\Phi S_{nw}>\xi P^{2}$, the mask surface and the bottom sections of the NW sidewalls are fully shadowed. No material exchange between the NWs and the substrate or between the neighboring NWs is possible. The upper sections of NWs are fed only by the direct vapor flux. This stage is identical for any NWs and is described by the growth law
$$\frac{d}{dt}\left[\pi R^{2}L\right]=\xi P^{2}v,$$
$$\frac{dL}{dt}=v\Phi +\frac{2\lambda_{f}}{R}v_{f}\Phi_{f}\;\mathrm{in}\;\mathrm{stage}\;2,$$
(15)$$\frac{dL}{dt}=\frac{2\lambda }{R}v\Phi \;\mathrm{in}\;\mathrm{stage}\;3.$$

According to these equations, the NW volume increases linearly with time. The NW length evolution is given by the same equation as in stages 2 or 3, depending on which stage precedes the asymptotic stage. Exact solutions to these equations in stage 2 for the axial growth are given in Ref. [46], with stage 3 being a simple limiting case. Pre-history of NW evolution does not influence the form of the asymptotic growth law but determines the initial conditions for the NW length and radius at the beginning of stage 4.

The durations of different growth stages and even their occurrence depend on many factors, including the growth temperature and pitch in a given epitaxy technique. For example, stage 3 should not occur in MBE growth with low desorption rates of group III atoms [45,46]. The asymptotic stage can actually start very early in dense NW arrays, such as MBE-grown arrays of InAs NWs of Ref. [29], whose growth kinetics were strongly influenced by the pitch from the very beginning. On the other hand, high-temperature NW growth should be less dependent on the array pitch. For example, the pitch dependence observed in SAG of GaAs NWs by MOVPE at 750 °C in Ref. [30] was weaker than in Ref. [29]. Same applies to MBE of self-induced GaN NWs on amorphous layers [11,12,13,14,15,48] or SAG GaN NWs on patterned SiO_x_/Si(111) substrates [21,22,23]. According to Equation (14), the pitch dependence of the axial and radial growth rates is determined by the factor $\Phi S_{nw}/\xi P^{2}$, which decreases for lower Φ and vanishes at Φ→0. Therefore, enhanced desorption at high temperatures may lead to an almost pitch-independent morphological evolution in stages 1, 2, and 3. Pitch-independent growth at high temperatures continues for a much longer time compared to the low-temperature growth at Φ≅1. The growth rate enhancement factor due to re-emitted species in the pitch-independent step is close to 2 for both axial and radial growths in VPE. In the directional MBE method, the radial growth rate also increases by a factor of 2. However, the axial growth rate is not enhanced due to a negligible secondary flux onto the NW top facet (the situation is different for VLS NWs whose apical droplets are exposed to re-emitted flux [45]).

## 3. Rigorous Approach Based on Diffusion Equations

The NW geometry considered in this section is shown in Figure 1d. As discussed above, untapered NW grows by incorporating group III adatoms to cylindrical shell layer progressing along the sidewalls and circular 2D island growing on the top facet. These steps are fed by adatoms coming from the “terraces” (with the surface concentrations n on the top facet and $n_{f}$ on the sidewalls) and the layer above (with the surface concentrations $n^{\prime }$ on top of 2D island and $n_{f}^{\prime }$ on top of the shell layer). We consider 2D island, which nucleates at the center of the top facet. However, the resulting diffusion flux remains the same if ring-shaped island nucleates at the facet edge and propagates to the center. Both steps evaporate adatoms at the rates determined by the effective equilibrium concentrations at the step edges [50]
(16)$$n_{*}=\frac{n_{3}^{eq}n_{5}^{eq}}{n_{5}},\;n_{f*}=\frac{n_{f3}^{eq}n_{f5}^{eq}}{n_{f5}}.$$

Here, $n_{3}^{eq}n_{5}^{eq}$ and $n_{f3}^{eq}n_{f5}^{eq}\;$ are the temperature-dependent equilibrium activities per III-V pair on the top and side facet, respectively. Under group V-rich conditions, the spatially uniform surface concentrations of group V adatoms $n_{5}$ and $n_{f5}$ are determined by the balance of the incoming atomic flux of group V adatoms and their desorption [50].

The surface concentration $n^{\prime }$ of a “separated” adatom population on top of 2D island is obtained from the stationary 2D diffusion equation
$$D\Delta n^{\prime }+\chi J-\frac{n^{\prime }}{\tau }=0,$$
(17)$${\left.\frac{dn^{\prime }}{d\rho }\right|}_{\rho =0}=0,\;n^{\prime }\left(\rho =r\right)=n_{*}.$$

Here, D is the diffusion coefficient on the top facet, J is the arrival rate of group III atoms from vapor (in nm^−2^s^−1^), τ is the adatom lifetime on the top facet before desorption, and χ is pyrolysis efficiency of a group III precursor in VPE or a geometrical coefficient related to the MBE beam angle φ according to χ=cosφ. The boundary conditions are well defined and independent of other adatom populations. The first boundary condition is due to symmetry at ρ=0. The second boundary condition requires that adatoms at the step edge (at ρ=r, with r as the island radius) are at equilibrium in solid. The diffusion flux into the island from the island top is given by
(18)$$j^{\prime }=-2\pi rD{\left.\frac{dn^{\prime }}{d\rho }\right|}_{\rho =r}.$$

This gives
(19)$$j^{\prime }=2\pi r\lambda \left(\chi J-\frac{n_{*}}{\tau }\right)\frac{I_{1}\left(r/\lambda \right)}{I_{0}\left(r/\lambda \right)},$$
where $\lambda =\sqrt{D\tau }$ is the diffusion length of group III adatoms on the top facet, and $I_{m}\left(r/\lambda \right)$ are the modified Bessel functions of the first type of order m.

The adatom population on the terrace between the island and the top edge is more complex. It is influenced by a diffusion flux coming from the NW sidewalls, which, in turn, depends on the adatom population $n_{f}$. Let us first calculate the diffusion flux entering 2D island from the outer ring for arbitrary diffusion flux from the NW sidewalls. The corresponding diffusion problem is given by
$$D\Delta n+\chi J-\frac{n}{\tau }=0,$$
(20)$$n\left(\rho =r\right)=n_{*},\;\lambda {\left.\frac{dn}{d\rho }\right|}_{\rho =R}=x.$$

The first boundary condition at the island periphery is the same as before. The x is related to the diffusion flux through the periphery 2πR. The solution is given by
(21)$$j=2\pi r\lambda \left(\chi J-\frac{n_{*}}{\tau }\right)y\left(\frac{r}{\lambda }\right)+x\frac{2\pi r\lambda }{\tau }\left[K_{1}\left(\frac{r}{\lambda }\right)-K_{0}\left(\frac{r}{\lambda }\right)y\right],$$
with
(22)$$y=\frac{I_{1}\left(R/\lambda \right)K_{1}\left(r/\lambda \right)-K_{1}\left(R/\lambda \right)I_{1}\left(r/\lambda \right)}{I_{1}\left(R/\lambda \right)K_{0}\left(r/\lambda \right)+K_{1}\left(R/\lambda \right)I_{0}\left(r/\lambda \right)}.$$

Here, $K_{m}\left(X\right)$ are the modified Bessel functions of the second type of order m.

The surface concentration of sidewall adatoms in the upper NW section between the step and the NW top is governed by the diffusion equation:$$D_{f}\frac{d^{2}n_{f}}{dz^{2}}+\chi_{f}J-\frac{n_{f}}{\tau_{f}}=0,$$
(23)$$\left(z=0\right)=n_{f*},\;n_{f}\left(z=L-H\right)=n_{f*}.$$

Here, the quantities $D_{f}$, $\chi_{f}$, and $\tau_{f}$ have the same meaning as in Equation (17) but for the NW sidewalls. In particular, χ=sinφ/π in MBE [44]. The first boundary condition at the adsorbing step edge is well defined and obvious. Conversely, the second boundary condition at the transparent upper edge of the NW can have different forms. The most general form requires the continuity of the diffusion flux across the edge and expresses it through the adatom transition rates from the NW sidewalls to its top and in the opposite direction [44,49,50]. Unfortunately, these transition rates are generally unknown. Here, we use the same boundary conditions for $n_{f}$ at the adsorbing and transparent edges, which corresponds to symmetric concentration profiles like in Ref. [54]. The unknown x is then obtained using
(24)$$\frac{\lambda }{\tau }x=D_{f}{\left.\frac{dn_{f}}{dz}\right|}_{z=0}=D{\left.\frac{dn}{d\rho }\right|}_{\rho =R},$$
which preserves the continuity of the diffusion flux. The result for x is given by
(25)$$x=\frac{\tau }{\lambda }\lambda_{f}\left(\chi_{f}J-\frac{n_{f*}}{\tau_{f}}\right)\frac{cosh\left(\frac{L-H}{\lambda_{f}}\right)-1}{sinh\left(\frac{L-H}{\lambda_{f}}\right)},$$
where $\lambda_{f}=\sqrt{D_{f}\tau_{f}}$ is the desorption-limited diffusion length on the NW sidewalls.

The surface concentration of the “separated” adatom population at the NW bottom is obtained from
$$\frac{d^{2}n_{f}^{\prime }}{dz^{2}}+\chi_{f}J-\frac{n_{f}^{\prime }}{\tau_{f}}=0,$$
(26)$$n_{f}^{\prime }\left(z=0\right)=n_{f*},\;n_{f}^{\prime }\left(z=H\right)=n_{f*}.$$

Here, we again use the same boundary conditions at the step edge and the bottom edge of the NW in contact with the mask surface. If surface diffusion on the mask were enabled, the second boundary condition would link this adatom population to a population of adatoms on the mask [44,49,50]. However, we assume that all group III adatoms arriving onto the mask are re-emitted, in which case the choice of “adsorbing” boundary condition seems plausible. The diffusion flux of adatoms from the bottom part of the NW into the sidewall step is calculated by
(27)$$j_{f}^{\prime }=-2\pi RD_{f}{\left.\frac{dn_{f}^{\prime }}{dz}\right|}_{z=L-H}.$$

Using Equation (25) for x, the diffusion flux from the periphery of the top facet into 2D island is obtained in the form:(28)$$j=2\pi r\lambda \left(\chi J-\frac{n_{*}}{\tau }\right)y+2\pi r\lambda_{f}\left(\chi_{f}J-\frac{n_{f*}}{\tau_{f}}\right)\left[K_{1}\left(\frac{r}{\lambda }\right)-K_{0}\left(\frac{r}{\lambda }\right)y\right]\frac{cosh\left(\frac{L-H}{\lambda_{f}}\right)-1}{sinh\left(\frac{L-H}{\lambda_{f}}\right)}.$$

The diffusion fluxes feeding the shell are obtained in the form:$$j_{f}=2\pi r\lambda_{f}\left(\chi_{f}J-\frac{n_{f*}}{\tau_{f}}\right)\frac{cosh\left(\frac{L-H}{\lambda_{f}}\right)-1}{sinh\left(\frac{L-H}{\lambda_{f}}\right)},$$
(29)$$j_{f}^{\prime }=2\pi r\lambda_{f}\left(\chi_{f}J-\frac{n_{f*}}{\tau_{f}}\right)\frac{cosh\left(\frac{H}{\lambda_{f}}\right)-1}{sinh\left(\frac{H}{\lambda_{f}}\right)}.$$

The total fluxes feeding 2D island and the sidewall shell are given by
(30)$$j_{tot}=j+j^{\prime },\;j_{f,tot}=j_{f}+j_{f}^{\prime },$$
with $j^{\prime }$ defined by Equation (19). Figure 2 shows the normalized diffusion flux into 2D island on the NW top, ${\tilde{j}}_{tot}=j_{tot}/\left[\pi R^{2}\left(\chi J-n_{*}/\tau \right)\right],$ as a function of its normalized radius r/λ, for small (Figure 2a) and large (Figure 2b) values of R/λ, at a fixed $\left[cosh\left(H/\lambda_{f}\right)-1\right]/sinh\left(H/\lambda_{f}\right)=1$. The curves were obtained from Equations (19) and (28)–(30). For thin NWs with R/λ≤0.25 in Figure 2a, the diffusion flux depends weakly on the island radius, meaning that all adatoms arriving from the NW sidewalls to the top incorporate into the island at any time regardless of its size. This assumption, used previously in Refs. [11,41,42], corresponds to the diffusion flux at r=R, shown by the dashed lines in both figures. For thick NWs with  R/λ≥1, the situation changes drastically. The diffusion flux is greatly reduced for small islands, meaning that a significant fraction of adatoms re-evaporate from the NW top and do not contribute to the axial growth. For very large  R/λ ratios, the upward diffusion of sidewalls adatoms may be present but has no influence on the axial growth rate. As expected, the diffusion-induced character of growth leads to faster axial growth rates of thinner NWs [11,18,25,33,43]. The axial NW growth rate increases by a factor ranging from ~10 to ~40 for thin NWs in Figure 2a and is lower than 3 for thick NWs in Figure 2b.

The diffusion fluxes given by Equations (19) and (28)–(30) depend not only on the NW radius R and length L but also on the radius of 2D island on the NW top r and the height of the sidewall facet H. This allows for a refined growth picture which includes the dynamics of step propagation. These fast processes do not influence, however, the mean NW growth rates. Therefore, we take an average of the diffusion fluxes over r and H to obtain the growth rates, which depend only on R and length L, similar to Ref. [50]. The mean growth rates are calculated using
(31)$$\frac{1}{\Omega_{35}}\pi R^{2}\frac{dL}{dt}=\frac{1}{LR}\int_{0}^{L}dH\int_{0}^{R}drj_{tot}\left(H,r\right),\;\frac{1}{\Omega_{35}}2\pi RL\frac{dR}{dt}=\frac{1}{L}\int_{0}^{L}dHj_{f,tot}\left(H\right).$$

Using Equations (19) and (28)–(30) and integrating them, we obtain the main result of this section:(32)$$\frac{dL}{dt}=2{\left(\frac{\lambda }{R}\right)}^{3}F\left(\frac{R}{\lambda }\right)v\Phi +\frac{L}{R}{\left(\frac{\lambda }{R}\right)}^{2}G\left(\frac{R}{\lambda }\right)\frac{dR}{dt},$$
(33)$$\frac{dR}{dt}=4{\left(\frac{\lambda_{f}}{L}\right)}^{2}ln\left[cosh\left(\frac{L}{2\lambda_{f}}\right)\right]v_{f}\Phi_{f}.$$

The parameters are given by
(34)$$v=\Omega_{35}\chi J,\;v_{f}=\Omega_{35}\chi_{f}J,\;\Phi =1-\frac{n_{*}}{\chi J\tau },\;\Phi_{f}=1-\frac{n_{f*}}{\chi_{f}J\tau_{f}}.$$

It is clear that v and $v_{f}$ are the arrival rates onto the top and sidewall facets, respectively, which are the same as in the empirical model of Section 2. The fractions of desorbed species, Φ and $\Phi_{f}$, are related to the effective equilibrium concentrations of adatoms on the two facets [33,50]. The steps grow at Φ>0 and decompose at Φ<0, with Φ=0 corresponding to the no-growth condition. These Φ have exactly the same meaning as in the empirical model. The functions F(R/λ) and G(R/λ) are defined according to
(35)$$F\left(\frac{R}{\lambda }\right)=\int_{0}^{R/\lambda }dxx\left[\frac{I_{1}\left(x\right)}{I_{0}\left(x\right)}+y\left(x\right)\right],\;G\left(\frac{R}{\lambda }\right)=\frac{1}{K_{1}(R/\lambda )}\int_{0}^{R/\lambda }dxx\left[K_{1}\left(x\right)-K_{0}\left(x\right)y\left(x\right)\right].$$

The axial and radial growth rates given by Equations (32) and (33) have a more complex form than Equation (1), obtained for the same geometrical model. It is interesting to note, however, that the only “geometrical” ratio L/R in Equation (32) for the axial growth rate is the same as in Equation (1). All other length and radius dependences in Equations (32) and (33) contain the ratios R/λ and $L/\lambda_{f}$. In the next section, we will show that the general growth obtained from the diffusion equations is reduced to the empirical models for stages 1, 2, and 3 in the limiting cases, while the complex functions of R/λ and $L/\lambda_{f}$ entering the general equations describe transitions between the limiting regimes.

## 4. Results and Discussion

For thin NWs with R/λ→0, the asymptotic behaviors of the modified Bessel functions in Equations (32) and (33) are given by $I_{0}\left(x\right)\to 1$, $I_{1}\left(x\right)\to x/2$, $K_{0}\left(x\right)\to -\ln\left(x/2\right)$, and $K_{1}\left(x\right)\to 1/x$. For thick NWs with R/λ→∞, we have $I_{m}\left(x\right)\to \exp\left(x\right)/\sqrt{2\pi x}$ and $K_{m}\left(x\right)\to \sqrt{\pi /2}\exp\left(-x\right)/\sqrt{x}$ for any m. Therefore, the axial NW growth rate given by Equation (32), which is reduced to
(36)$$\frac{dL}{dt}=v\Phi +\frac{L}{R}\frac{dR}{dt},\;\frac{R}{\lambda }\ll 1,\;\frac{dL}{dt}=\frac{2\lambda }{R}v\Phi ,\;\frac{R}{\lambda }\gg 1$$
for thin and thick NWs, respectively. For short NWs with $L/\left(2\lambda_{f}\right)\to 0$, $\ln\left[\cosh\left(x\right)\right]\mathrm{tends}\;\mathrm{to}\;x^{2}/2$. For long NWs with $L\left(2\lambda_{f}\right)\to \infty$, $\ln\left[\cosh\left(x\right)\right]\mathrm{tends}\;\mathrm{to}\;x$. Equation (33) for the radial NW growth rate becomes
(37)$$\frac{dR}{dt}=\frac{v_{f}\Phi_{f}}{2},\;\frac{L}{2\lambda_{f}}\ll 1,\;\frac{dR}{dt}=\frac{2\lambda_{f}}{L}v_{f}\Phi_{f},\;\frac{L}{2\lambda_{f}}\gg 1$$
for short and long NWs, respectively. Based on these equations, the limiting NW growth regimes considered in the empirical model are resumed. Stage 1 occurs for short and thin NWs with $L/2\lambda_{f}\ll$1 and R/λ≪1, where the morphological evolution is given by Equation (3). Stage 2 occurs for long enough NWs whose radius remains much smaller than the adatom diffusion length on the top facet, corresponding to long and thin NWs with $L/2\lambda_{f}\gg$1 and R/λ≪1. This stage is described by Equation (5). For long and thick NWs with $L/2\lambda_{f}\gg$1 and R/λ≫1, the growth is rendered in Stage 3, described by Equation (7). Therefore, the empirical and rigorous approaches are consistent and give the same results in the limiting cases.

We now consider the solutions to the governing equations describing the evolution of NW length and diameter in different stages. The NW growth is assumed independent of pitch, corresponding to $\Phi S_{nw}/\xi P^{2}\ll 1$ in Equation (14). Otherwise, the growth equations can only be solved numerically. At $\Phi S_{nw}/\xi P^{2}\ll 1$, the re-emitted group III atoms modify only the coefficients χ and $\chi_{f}$ in Equations (3), (5) and (7) in stages 1, 2, and 3, respectively. In stage 1, the NW radius increases linearly with time according to Equation (3) for dR/dt:(38)$$R=R_{0}+\frac{v_{f}\Phi_{f}}{2}t.$$

Using this in Equation (3) for dL/dt and integrating it, we obtain the length evolution in the form:(39)$$L=\alpha Rln\left(\frac{R}{R_{0}}\right),\;\alpha =2\frac{v\Phi }{v_{f}\Phi_{f}}=2\frac{\chi }{\chi_{f}}\frac{\Phi }{\Phi_{f}}.$$

This can equivalently be presented as a function of time using Equation (38). The NW length increases super-linearly with its radius or time because it collects adatoms from the length L/4, which increases with time. The most important parameter of our theory, α, equals twice the ratio of the equivalent 2D growth rates on the top and side NW facets. The NW length evolves faster for larger α. According to Equation (39), α is proportional to the ratio of supersaturations on the top and side facets, $\Phi /\Phi_{f}$, which is expected to be larger than unity in most cases to ensure anisotropic growth of high aspect ratio NWs. On the other hand, it tends to unite at high supersaturations corresponding to negligible desorption of group III atoms from the NW sidewalls or top, as often occurs in MBE growth of III-V NWs [33,45,46,55]. Another important parameter influencing the NW elongation is the geometric ratio $\chi /\chi_{f}$. As discussed above, the re-emitted group III species increase both axial and axial VPE growth rates by a factor of 2, which cancels in the $\chi /\chi_{f}$. Therefore, the minimum value of α for VPE at $\Phi /\Phi_{f}≅1$ and $\chi /\chi_{f}≅1$ is around 2. In the directional MBE method, the ratio $\chi /\chi_{f}$ contains the geometrical factor π. On the other hand, group III atoms re-emitted from the mask land on the NW sidewalls but not on the flat top facet, which is why $\chi /\chi_{f}$ ≅1/2 in the initial growth step. At $\Phi /\Phi_{f}≅1$, the minimum value of α for MPE is around π and should be within the range between π and 2π if some material is adsorbed by parasitic structures on the mask surface [22,23] of self-induced GaN islands nucleating concomitantly with NW growth [12,13,41]. From these considerations, the average length of NWs should increase faster in MBE than in MOVPE. The top (111) facet usually grows faster than the side (110) facets, meaning that $\Phi >\Phi_{f}$ or even $\Phi \gg \Phi_{f}$. This increases the values of α and enhances the axial growth with respect to radial. Without any radial growth, the NW length would increase exponentially with time from Equation (3) for dL/dt. Exponential growth mode was indeed observed for different Au-catalyzed VLS III-V NWs grown by MBE [35,55] and MOVPE [56]. More complex behaviors of the NW lengths, as studied in Refs. [46,47], are due to the additional contribution of the re-emitted flux and predict even faster axial growth. In catalyst-free growth, however, the radial growth is more pronounced and significantly reduces the rate of NW elongation.

The specific growth law given by Equation (39) is related to the choice of the a and b factors in Equation (2), according to which the NW sidewalls collect adatoms from the NW length, which is exactly twice larger than the collection length for the axial growth. At arbitrary a and b, the NW radius in stage 1 is also linear in time, while the NW elongation is described by the power-law dependence obtained previously in Ref. [40]:(40)$$L=R_{c}\left[{\left(\frac{R}{R_{0}}\right)}^{\gamma }-\frac{R}{R_{0}}\right],\;\gamma =\frac{2a}{b},\;R_{c}=\frac{\alpha R_{0}}{\gamma -1}.$$

This two-parametric growth law (with γ and α influencing differently the NW shape) allows for a vast variety of morphologies, from very long and thin NWs at large γ≫1 to flat NWs which rapidly become nanodiscs at γ<1, as observed in InAsSb NWs at high Sb contents [57].

The specific choice of a and b in the empirical approach is confirmed by the exact solutions to the diffusion equations, but only for the symmetric boundary conditions given by Equations (23) and (26). The values of a=1/4 and b=1/2 follow from the shapes of the adatom concentration profiles on the NW sidewalls, given by
$$\frac{n_{f}^{\prime }}{\chi_{f}J\tau_{f}}=1-\Phi_{f}cosh\left(\frac{z}{\lambda_{f}}\right)+\Phi_{f}\left[cosh\left(\frac{H}{\lambda_{f}}\right)-1\right]\frac{sinh\left(z/\lambda_{f}\right)}{sinh\left(H/\lambda_{f}\right)},\;0\leq z\leq H,$$
(41)$$\frac{n_{f}}{\chi_{f}J\tau_{f}}=1-\Phi_{f}cosh\left(\frac{z}{\lambda_{f}}\right)+\Phi_{f}\left[cosh\left(\frac{L-H}{\lambda_{f}}\right)-1\right]\frac{sinh\left(z/\lambda_{f}\right)}{\sinh\left[\left(L-H\right)/\lambda_{f}\right]},\;0\leq z\leq L-H.$$

These shapes are universal in terms of the dimensionless coordinate $z/\lambda_{f}$, with $\Phi_{f}$ affecting only the heights of the maxima but not their positions and $L/\lambda_{f}$ not affecting the fractions of adatoms used for radial and axial growths. Figure 3 shows the dimensionless adatom concentrations versus $z/\lambda_{f}$ for $H/\lambda_{f}=$ 0.3, 0.5, and 0.7. At $H/\lambda_{f}=$ 0.3, the step is closer to the NW base and rapidly consumes adatoms from the bottom part of the NW, which is why the adatom population in the upper part is denser. At $H/\lambda_{f}=$ 0.5, the step is right at the middle of the NW, and the adatom concentration profiles are symmetric. At $H/\lambda_{f}=$ 0.7, the profiles are reversed with respect to $H/\lambda_{f}=$ 0.3. The fraction of adatoms collected by the sidewall step equals 1/2, on average, and the fraction of adatoms collected by the NW top equals 1/4, on average. This result is exact at $H/\lambda_{f}=$ 0.5. Therefore, the empirical choice of a and b values follows from the symmetric boundary conditions for the sidewall population of adatoms at the NW top, base, and middle. This symmetry may be broken in more general boundary conditions employing the transition rate constants [44,49,50] and in a more complex picture of radial growth involving multiple steps and tapered shells [53]. This important question requires a separate study and will be considered elsewhere. However, the choice of parameters determining the growth kinetics and morphology seems plausible for untapered NWs and was demonstrated by two different methods.

The solutions in stage 2 are obtained as follows. Dividing dL/dt to dR/dt given by Equation (5) gives the NW elongation as a function of R:(42)$$\frac{dL}{dR}=\left(\frac{\alpha }{4\lambda_{f}}+\frac{1}{R}\right)L.$$

Upon integration, we obtain the NW morphology $L\left(R\right)$ in the form:(43)$$L=L_{1}\frac{R}{R_{1}}exp\left[\frac{\alpha }{4}\frac{\left(R-R_{1}\right)}{\lambda_{f}}\right].$$

Here, $L_{1}≅4\lambda_{f}$ and $R_{1}$ are the NW length and radius at the end of stage 1. To find the time-dependent radius, we use Equation (43) in Equation (5) for dR/dt. This gives a differential equation:(44)$$\frac{d}{dt}\left(\frac{R}{\lambda_{f}}\right)=\frac{2R_{1}v_{f}\Phi_{f}}{L_{1}\lambda_{f}}exp\left(\frac{\alpha }{4}\frac{R_{1}}{\lambda_{f}}\right)\frac{1}{\left(R/\lambda_{f}\right)}exp\left(-\frac{\alpha }{4}\frac{R}{\lambda_{f}}\right).$$

Integrating them, we obtain
(45)$$R=\frac{4\lambda_{f}}{\alpha }\left\{1+W\left[\exp\left(\frac{\alpha R_{1}}{4\lambda_{f}}-1\right)\left(\frac{\alpha R_{1}}{4\lambda_{f}}-1+\frac{t-t_{1}}{T_{2}}\right)\right]\right\},$$
with the characteristic time constant
(46)$$\frac{1}{T_{2}}={\left(\frac{\alpha }{2}\right)}^{2}\frac{2R_{1}v_{f}\Phi_{f}}{L_{1}\lambda_{f}}.$$

The Lambert function $W\left(Y\right)$ is defined as the root of an equation $W\exp\left(W\right)=Y$. These growth laws, determined by the parameters α and $\lambda_{f}$, are more complex than the linear elongation law at a constant radius with the inverse length–radius correlation, as often considered in the growth modeling of VLS [33] and catalyst-free [11,43] NWs.

In stage 3, the length evolution as a function of R following Equation (7) is determined by
(47)$$\frac{dL}{dR}=\beta \frac{L}{R},\;\beta =\frac{\lambda }{2\lambda_{f}}\alpha .$$

Integration gives
(48)$$L=L_{2}{\left(\frac{R}{R_{2}}\right)}^{\beta },$$
where $L_{2}$ and $R_{2}$ are the NW length and radius at the end of stage 2. The time evolution of the NW morphology is readily obtained by the integration of Equation (7) for dR/dt with this L. Both length and radius increase sub-linearly with time
(49)$$L=L_{2}{\left[1+\left(\beta +1\right)\frac{\left(t-t_{2}\right)}{T_{3}}\right]}^{\beta /\left(\beta +1\right)},\;R=R_{2}{\left[1+\left(\beta +1\right)\frac{\left(t-t_{2}\right)}{T_{3}}\right]}^{1/\left(\beta +1\right)}.$$

Here, the time constant T_3_ is defined according to 1⁄T_3_ = 2λ_f_v_f_Φ_f_/(L_2_R_2_).The length shows a scaling power-law dependence on its radius according to Equation (48). This growth behavior was earlier predicted in Refs. [40,42]. The value of β equals α/2 at $\lambda =\lambda_{f}$ and hence may be close to unity in VPE. At β=1, the power exponents in Equation (49) for length and radius are the same and equal 1/2. The NW length can increase with time even slower than its radius when $\lambda <\lambda_{f}$, which would result in a decreasing aspect ratio. Therefore, the fabrication of high aspect ratio NWs with very large radii may be very difficult. One important property, which follows directly from Equation (7), is the linear increase of the NW sidewall area with time:(50)$$2\pi RL=2\pi R_{2}L_{2}\left[1+\left(\beta +1\right)\frac{\left(t-t_{2}\right)}{T_{3}}\right].$$

The NW aspect ratio in stage 3 is usually much larger than unity. Therefore, the sidewall surface area approximately equals the total NW surface area $2\pi RL+\pi R^{2}$, whose time evolution is also close to linear.

Figure 4 shows the time evolution of the NW morphology starting from the same initial radius of 50 nm with the same growth rate on the sidewalls $v_{f}\Phi_{f}=$2 nm/min, at a fixed diffusion length on the NW sidewalls of 125 nm, for 2 different α= 2 and 2π. The curves were obtained from Equations (38) and (39) in stage 1, (43), (45) and (46) in stage 2, and (49) in stage 3. The 2 values of α correspond to the pitch-independent VPE and MBE growths at $\Phi =\Phi_{f}=1$, respectively, according to the discussion above. The NW elongation is faster, and the radial growth is suppressed for larger α in all stages. At a short $\lambda_{f}$ of 125 nm, super-linear behavior of the NW lengths with time is observed only for short NWs with lengths below ~500 nm. The critical NW radius for the transition from stage 2 to stage 3 is chosen at 300 nm. This radius is not reached by the NW at α=2π in 300 min of growth. At α=2, the NW is thicker from the very beginning, which is why stage 3 begins early (at a length of around 1250 nm), and the axial growth becomes more sub-linear. The total surface area of both NWs reaches the linear asymptote after a certain time. The NW volume increases super-linearly in both cases. This example shows that α is the critical parameter for obtaining high aspect ratio NWs in each stage.

The length-radius dependence of catalyst-free GaN [11] and different III-V [24,25,26,27,28] NWs after a given growth time shows an inverse correlation (usually with $R^{-1}$ or $R^{-2}$ radius-dependent terms), which are well known for diffusion-induced NW growth [33,34,35,43,44,55]. We note, however, that applying the models originally developed for NWs growing at a constant radius (which is a parameter of a growth equation for length) [33,34,44] to NWs whose radius increases from the very beginning (and hence becomes a function of time similarly to length) requires some care. Strictly speaking, this is valid only at negligible radial growth. Otherwise, one should consider an NW ensemble with radii $R\left(t\right)$, which have started from different $R_{0}$ at zero time and become thicker due to the radial growth. These NWs have reached different lengths $L\left(t\right)$ by the same moment of time t because their radii were different at all moments of time before. The resulting $L\left(R\right)$ dependence may be very different from the predictions of any model at a time-independent $R=R_{0}$. The spread in the observed NW radii is due to different $R_{0}$ from which they start, but the measured range of $R\left(t\right)$ can be very far from the initial range of $R_{0}$. This is clearly seen, for example, in the length and radius histograms of self-induced GaN NWs given in Ref. [11], where the NW radii gradually increased with time. Modeling $L\left(R\right)$ correlations at a given time for NWs with the radial growth is complicated by the fact that NWs starting from different initial radii $R_{0}$ have different pre-history (in particular, different critical times corresponding to transitions from one growth stage to the other). Furthermore, the measured NW radii after growth give no indication of their initial radii at zero time, unless they were measured before or at the very beginning of the NW growth [12,13,29,30,41,42]. Figure 5 shows an example of length–radius correlations for NWs with the same parameters as in Figure 4, starting from different initial radii $R_{0}$, after different NW growth times. A growth time of 28 min corresponds to the end of stage 1 for NWs at α=2π. Therefore, all NWs shown in the figure are in stage 1, with lengths and radii given by Equations (38) and (39). This corresponds to the length–radius correlation of the form $L=\alpha Rln[R/(R-v_{f}\Phi_{f}t/2)]$, which is very different from the $R^{-1}$ or $R^{-2}$ radius-dependent terms discussed earlier for catalyst-free NWs [11,24,25,26,27,28]. The curve $L=\left[A+{\left(\Lambda /R\right)}^{2}\right]H$, at a constant radius, is close to our curve at A= 0.5, Λ= 90 nm (with H=352 nm as the 2D layer thickness at 28 min), if fitted over a limited range. However, the origins of the two dependencies are principally different. These considerations are not specific to catalyst-free NWs and apply equally to VLS NWs whose elongation can be influenced by radial growth [35,46].

## 5. Theory and Experiment

SAG GaAs NWs of Ref. [30] were obtained by MOVPE at 750 °C on patterned SiO_2_/GaAs substrates, with a 600 nm pitch and the variable nominal diameter of circular openings (pores) from 125 to 225 nm. The NW shape was hexahedral without any noticeable tapering from base to top. Figure 6a–c shows the measured average lengths, radii, and volume of NWs grown for different times up to 80 min in differently sized pores. Figure 6d shows the evolution of the length–radius dependencies after different growth times from 20 to 80 min. The fits for the NW lengths and radii in Figure 6a,b were obtained using Equations (38) and (39) in stage 1 and Equations (43), (45) and (46) in stage 2, with the parameters listed in Table 1. The NW volume in Figure 6c was calculated for hexahedral geometry. The length-radius correlations in Figure 6d were obtained using the calculated NW length and radii in the arrays of differently sized pores. The kinetic data features the general trends discussed above. The NW lengths increase super-linearly with time in the beginning and then become slightly sublinear. The NW radii increase sub-linearly with time. The NW volume increases super-linearly with time. This increase is slower than the exponential. The NW length is larger in arrays of smaller pores at any time, which confirms the diffusion-induced character of growth. The fits require very high values of α in the range from 10.5 to 14.1, suggesting that the radial growth is strongly suppressed by a lower effective supersaturation at the NW sidewall facets compared to the top. This enables highly anisotropic NWs with relatively large radii. The transition from super-linear to slightly sub-linear length evolution is observed at a length $L_{1}$ of about 2000 nm in all cases. We used the growth equations at $L_{1}=4\lambda_{f}$. Therefore, the diffusion length of Ga adatoms on the NW sidewalls $\lambda_{f}$ is estimated at 500 nm. The NW radii at the end of growth are well below this value. Assuming $\lambda ≅\lambda_{f}$ for the Ga diffusion length on the top facet, stage 3 should not start for the growth times employed in the experiments. As a result, no transition is observed to a more pronounced sub-linear behavior of the NW lengths.

GaN NWs of Ref. [42] were grown by plasma-assisted MBE on Si(111) substrates covered with a 2 nm thick Si_x_N_y_ interlayer, as described in detail in Refs. [12,13]. The growth proceeded under N-rich conditions at 780 °C, with a Ga deposition rate of 2.7 nm/min. Anisotropic NWs formed from the Volmer–Weber spherical cap islands at a critical island radius of ~5 nm via the self-induced mechanism [12,13,41]. The actual NW growth time was monitored using reflection high-energy electron diffraction and varied between 4.5 min and 379.5 min. The NW morphology was hexahedral. Figure 7 shows the evolution of NW morphology with time. In Refs. [41,42], it was noticed that the average NW length scaled approximately as a power law of its average radius at the same moment of time, which is why the NW growth kinetics was fitted by Equations (48) and (49) with β=2.46. This yields a super-linear increase of the NW length with its radius and sub-linear time evolution of the NW length and radius, as in our stage 3. We note, however, that neglecting the direct impingement of Ga atoms on the top NW facet relative to surface diffusion from the side facets was essential for obtaining the scaling growth law in the kinetic model of Ref. [42].

The solid lines in Figure 7 show the fits obtained from Equations (38) and (39) in stage 1 and Equations (43), (45) and (46) in stage 2, with the parameters summarized in Table 2. The fit of the NW length evolution with its radius is not significantly different from Ref. [42]. However, the other curves obtained within the modified model fit the data better. In particular, the power-law model largely underestimates the NW length for the longest NW growth time of 379.5 min in Figure 7b. This model yields a linear increase of the NW surface area with time, as discussed above. This contradicts the data showing a super-linear increase in the surface area, which is better fitted by the modified model. Larger deviations of the fitting curves from the datapoints after longer growth times are probably explained by a complex material exchange between the neighboring NWs [38], which is not taken into account in our model for isolated NW. Figure 8 shows the NW length and radius versus time on a logarithmic scale, demonstrating that the fits at short NW growth times are also better within the modified model. This is related to the fact that the modified model predicts a linear increase of the NW radius in the short stage 1 before reaching an NW length of ~$4\lambda_{f}$, with the corresponding super-linear increase of the length according to Equation (39). This is followed by the sub-linear evolution of the length and radius in stage 2. The power-law model predicts sub-linear behaviors of the length and radius at any time. Regarding the parameters in quo 2, the fitting value of α=7.1 is lower than for MOVPE GaAs NWs in Figure 6. However, the initial radius of GaN islands is very small, in the order of only 5 nm, which enables highly anisotropic GaN NWs with an extremely high aspect ratio L/R of about 45 at the end of growth. Both $v_{f}\Phi_{f}$ and vΦ values are noticeably lower than the Ga-limited deposition rate of 2.7 nm/min. With a Ga beam angle of 21° [42] and assuming that the desorbed Ga atoms land on the NW sidewalls but not on their tops, the desorption factors are estimated at Φ=0.61 and $\Phi_{f}=0.70$. They are similar for the NW side and top facets. The diffusion length of Ga adatoms on the NW sidewalls is estimated at 35 nm, which is close to the previously reported values of ~40 nm [11,14,43].

## 6. Conclusions

The empirical approach to modeling the growth chronology of untapered catalyst-free III-V and III-nitride NWs reveals three distinct stages of NW growth. In the first stage, adatoms from one-half of the NW length contribute to the radial growth, and adatoms from one-quarter of the NW length are used for the axial growth. In the second stage, the sidewall step collects adatoms from a length of $2\lambda_{f}$ around the step, while the NW top collects adatoms from length $\lambda_{f}$. When the NW radius grows larger than the adatom diffusion length λ on the top facet, the NW axial growth is limited by this λ and can no longer be enhanced by adatoms diffusing from the NW sidewalls. The rigorous approach based on the stationary diffusion equations for the four populations of adatoms on each side of the side and top steps provides a refined growth picture. It is reduced to the three growth stages in the limiting regimes, which depend on the NW geometry and growth conditions. The model applies for untapered NWs grown at relatively high temperatures such that the adatom diffusion lengths are limited by desorption rather than nucleation of other steps. The NW growth can be influenced by the pitch or surface density of self-induced islands at any stage. However, the pitch dependence becomes weaker for higher desorption rates of group III adatoms. An important relationship is established between the NW growth laws and the boundary conditions for the adatom concentrations at the NW top and bottom. The developed approach presents the first attempt at the rigorous modeling of catalyst-free NW growth, similar to what has been achieved for VLS NWs. The model fits quite well the data on the morphological evolution of SAE GaAs NWs and self-induced GaN NWs growth by different epitaxy techniques.

Several important aspects of catalyst-free NW growth remain to be studied. In particular, the growth theory should be extended to more complex geometry, including tapered NWs with multilayer growth on the sidewalls, with a possibility of step bunching. This can be performed by considering an echelon of steps on the NW sidewalls with the initial separation determined by the nucleation probability at the NW base. Different types of boundary conditions to the diffusion equations should be carefully investigated versus the growth parameters, material constants, and NW morphologies. In particular, the Kramers-type boundary condition employing the transition rate constants over the transparent step at the NW top [44,49] should be tested, leading to asymmetric adatom concentration profiles. The effect of doping on the growth behavior, which is well known for 2D layered materials [58] and III-V NWs [59], should be considered in detail. The pitch-dependent NW growth kinetics, as described in Ref. [29], should be studied using the general growth laws given by Equations (13)–(15), with pitch-dependent re-emitted flux. The model can be applied to modeling the composition of III-V and III-nitride ternary NWs and related nanostructures based on a group III intermix, with different diffusivities and equilibrium concentrations on different surfaces. The vapor–solid distributions connecting the solid composition in a III-V NW with the vapor composition can be obtained using the expressions for the diffusion fluxes of different group III adatoms entering the step on the NW top. We plan to study these questions in a forthcoming work and consider more experimental data for catalyst-free III-V and III-nitride NWs from the viewpoint of the obtained results. Overall, the developed approach should be useful for understanding the growth kinetics and controlling the NW morphology by tuning the growth parameters and template geometries.

## Figures and Tables

**Figure 1 nanomaterials-13-01253-f001:**
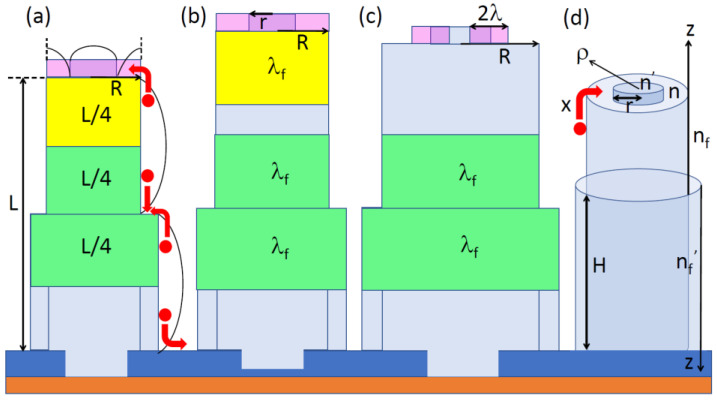
Illustration of (**a**) stage 1 for short and thin NW, (**b**) stage 2 for long and thin NW, and (**c**) stage 3 for long and thick NW. The pink zones show the areas of the top facet from which group III adatoms enter circular island growing on the NW top. The yellow zones show the areas of the side facets contributing to the axial growth by attachment to the island on top. The green zones show the fractions of the side facets from which group III adatoms are collected by the single-step propagating on the NW sidewalls. In stage 1, the NW collects adatoms from the whole top facet and the upper section of the sidewalls of height L/4. Radial shell collects adatoms from the height L/2. The curved lines show symmetric adatom concentration profiles on the NW sidewalls and at the top. In stage 2, the sidewall collection length is reduced to $\lambda_{f}$. In stage 3, no sidewall adatom can reach the step growing on top, whereas the top collection area is reduced to 2πrλ. (**d**) Illustration of cylindrical NW geometry and the four populations of adatoms on top of 2D island ($n^{\prime }$), on the top terrace (n), on top of cylindrical shell around the NW core ($n_{f}^{\prime }$), and on the terrace $n_{f}$. x denotes the diffusion flux from the NW sidewalls to its top. The ρ and z axes show the directions chosen in the corresponding diffusion equations. The red arrows in (**a**,**d**) show the directions of the adatom diffusion fluxes.

**Figure 2 nanomaterials-13-01253-f002:**
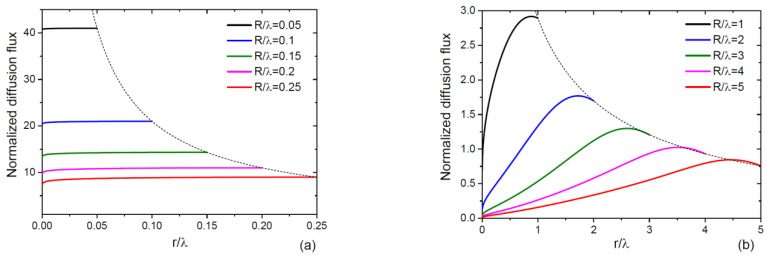
Normalized diffusion flux into 2D island on the NW top versus its non-dimensional radius r/λ for different R/λ between (**a**) 0.05 and 0.25, and (**b**) 1 and 5. The dashed lines show the fluxes at r=R. The fluxes are almost independent of the island radius for “thin NWs” in (**a**), and highly dependent on the radius for “thick” NWs in (**b**). The axial growth rate enhancement due to surface diffusion reaches a factor of 41 for the thinnest NWs with R/λ=0.05 in (**a**) and is lower than 3 for thicker NWs with R/λ≥1 in (**b**).

**Figure 3 nanomaterials-13-01253-f003:**
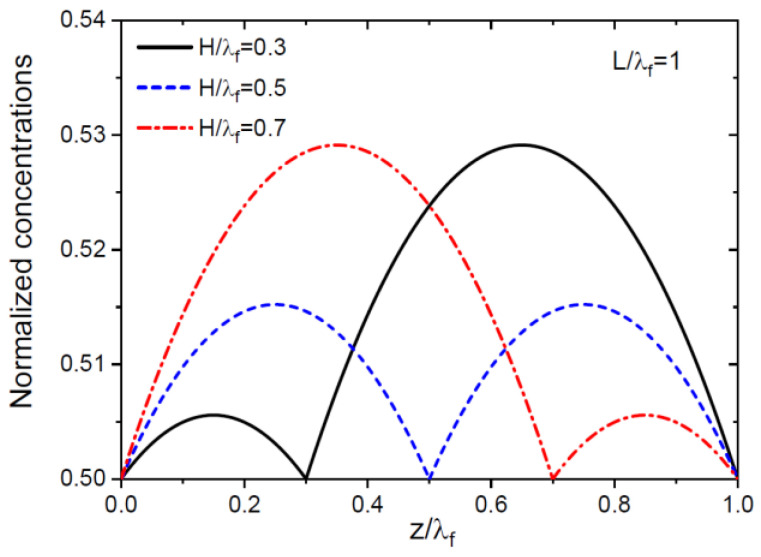
Dimensionless adatom concentrations on the NW sidewalls versus the normalized height along the NW axis for different $H/\lambda_{f}$ shown in the legend and obtained from Equation (41) at $L/\lambda_{f}=1$.

**Figure 4 nanomaterials-13-01253-f004:**
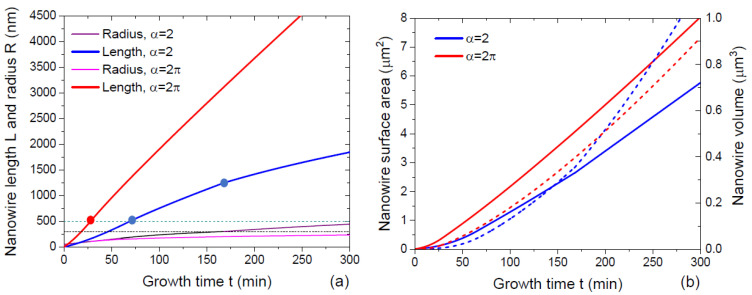
Morphological evolution of NWs in terms of (**a**) length and radius, and (**b**) total surface area and volume, for two different α of 2 and 2π. Horizontal dashed lines in (**a**) correspond to the length $L_{1}=$ 500 nm, at which the growth is transitioned from stage 1 to stage 2, and to the radius $R_{2}=$ 300 nm, at which stage 3 starts. Solid and dashed lines in (**b**) show the NW surface areas and volumes, respectively. Circles in (**a**) show the transitions from stage 1 to stage 2 and from stage 2 to stage 3 at α= 2. The NW length at α=2π is systematically larger, and the radius is systematically lower than at α=2. The length evolution is super-linear in stage 1, slightly sub-linear in stage 2, and markedly sublinear in stage 3. The time evolution of the NW surface area becomes almost linear in the large time limit. The NW volume increases super-linearly with time in all stages. The NW at α= 2 is shorter and thicker than at α=2π, which is why the NW volumes are similar. In the large time limit, the NW volume increases even faster at α= 2.

**Figure 5 nanomaterials-13-01253-f005:**
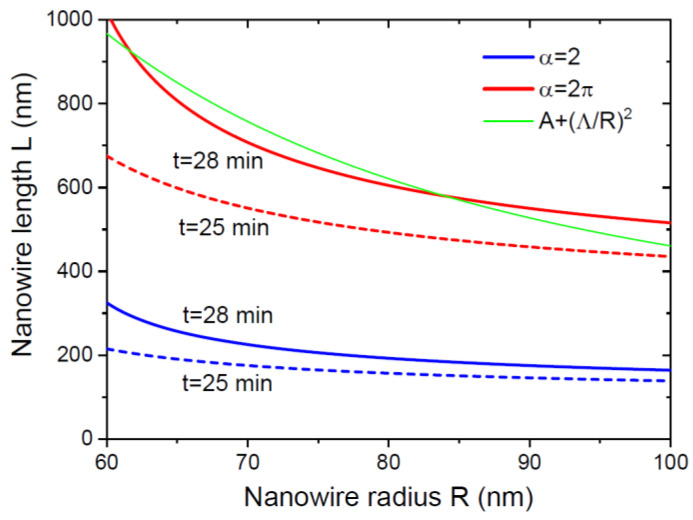
Length–radius correlations of the same NWs as in Figure 4 after different growth times of 25 min (dashed lines) and 28 min (solid lines). A growth time of 28 min corresponds to the end of stage 1 at α=2π. The green line shows the diffusion-induced curve at a constant radius, given by $L=\left[A+{\left(\Lambda /R\right)}^{2}\right]H$, with H= 352 nm as the equivalent 2D layer thickness after 28 min at α=2π.

**Figure 6 nanomaterials-13-01253-f006:**
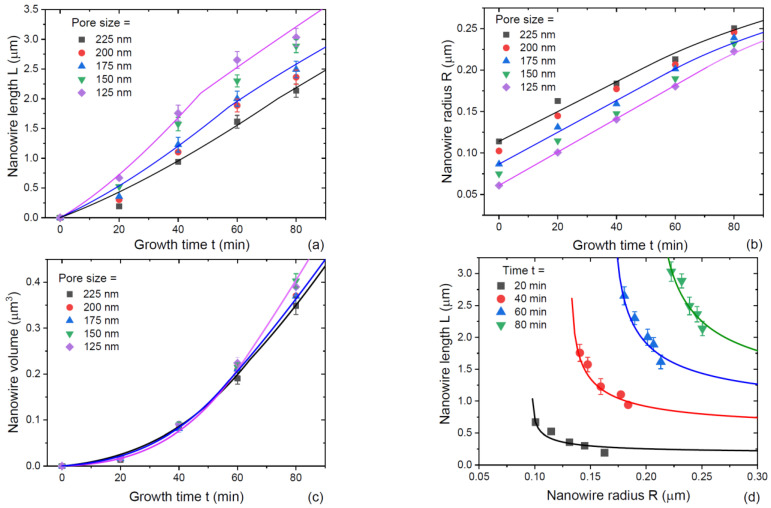
Evolution of the morphology of SAG GaAs NWs grown by MOVPE at 750 °C with different nominal pore diameters shown in the legends: (**a**) length, (**b**) radius, (**c**) volume, and (**d**) length–radius dependencies at different times. Symbols represent the data from Ref. [30]. Lines are the fits obtained within the model with the parameters summarized in Table 1.

**Figure 7 nanomaterials-13-01253-f007:**
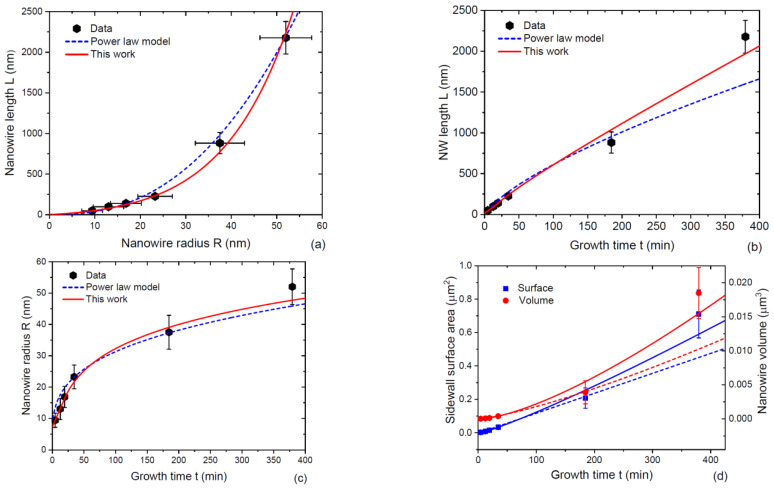
Evolution of the morphology of self-induced GaN NWs grown by MBE at 780 °C: (**a**) length versus radius, (**b**) length versus the NW growth time, (**c**) radius versus the NW growth time, and (**d**) volume versus the NW growth time. Symbols represent the data from Ref. [42]. The dashed lines are the fits by the scaling power-law model from Ref. [42], with a power exponent of 2.46. The solid lines are the fits obtained within the modified model of this work with the parameters summarized in Table 2.

**Figure 8 nanomaterials-13-01253-f008:**
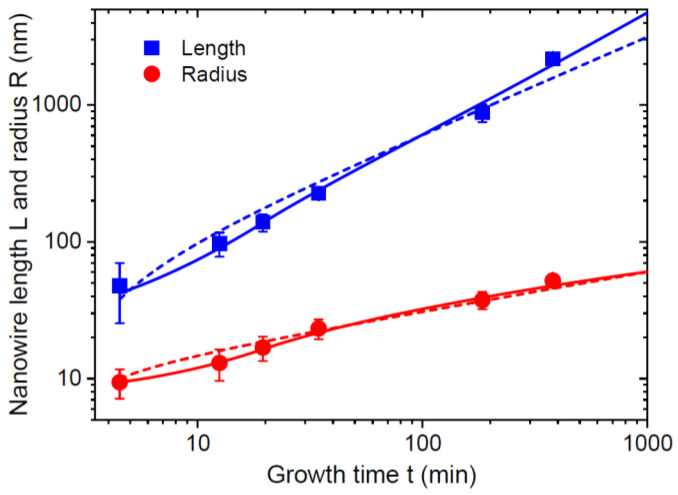
Evolution of the length and radius of self-induced GaN NWs in logarithmic scale. Symbols represent the data from Ref. [42]. The dashed lines are the fits by the scaling power-law model from Ref. [42]. The solid lines are the fits obtained within the modified model.

**Table 1 nanomaterials-13-01253-t001:** Fitting parameters for SAG GaAs NWs grown by MOVPE.

Pore Diameter (nm)	$2D\;\mathrm{Growth}\;\mathrm{Rate}\;\mathrm{on}\;\mathrm{the}\;\mathrm{Sidewalls}\;v_{f}\Phi_{f}$ (nm/min)	α	$\mathrm{NW}\;\mathrm{Length}\;\mathrm{at}\;\mathrm{Transition}\;\mathrm{to}\;\mathrm{Stage}\;2\;L_{1}$ (nm)
125	4.03	14.1	2090
175	3.84	11.5	1855
225	3.60	10.5	2005

**Table 2 nanomaterials-13-01253-t002:** Fitting parameters for self-induced GaN NWs grown by MBE.

$2D\;\mathrm{Growth}\;\mathrm{Rate}\;\mathrm{on}\;\mathrm{the}\;\mathrm{Sidewalls}\;v_{f}\Phi_{f}$ (nm/min)	α	2D Growth Rate on the Top vΦ (nm/min)	$\mathrm{NW}\;\mathrm{Length}\;\mathrm{at}\;\mathrm{Transition}\;\mathrm{to}\;\mathrm{Stage}\;2\;L_{1}$ (nm)	$\mathrm{Diffusion}\;\mathrm{Length}\;\mathrm{on}\;\mathrm{the}\;\mathrm{Sidewalls}\;\lambda_{f}$ (nm)
0.93	7.1	1.65	140	35

## Data Availability

Not applicable.

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
