# Peer review of "Modeling Catalyst-Free Growth of III-V Nanowires: Empirical and Rigorous Approaches"

_nanomaterials, 2023, doi:10.3390/nano13071253_

Round 1

Reviewer 1 Report

1.First of all, it is necessary to clarify the significance of studying the growth model of GaN without catalyst. Secondly, what is the problem with growth with catalyst compared to growth without catalyst. Thirdly, can your proposed theory promote catalyst-free growth and solve the current problems of catalyst-free growth.

2. In line 89 of the text,Is there evidence to prove that Some atoms impinging onto the top facet will also be lost for NW growth.

3.In Figure7 (b) and (c),Why is it that the fitting is very close to the experimental data in a short period of time, but it will be very different when the time becomes larger.

4.In Figure 5, the relationship between length and diameter of nanowires under three different growth time conditions is given. Can the relationship under more growth time conditions be provided? in addition, what is the basis for selecting the growth time of 25min, 28min and 29min?

Author Response

The response letter is attached.

Reviewer 2 Report

This paper reported an empirical approach for modeling the growth of growths of un-tapered catalyst-free III-V nanowires, which fits quite well the data on the morphological evolution of SAE GaAs NWs and self-induced GaN NWs growth by different epitaxy techniques. I recommend that this paper can be accepted with minor revisions. Some comments are as follows:

1. Figure 1 shows the key models for illustrating three stages for nanowires with different lengths and thicknesses, but the content presented is vague and monotonous. It will be better to show the traces of adatoms in different stages. And the model of 2D islands is mentioned in the rigorous approach based on diffusion equations, which can be inserted in this figure.

2. In the manuscript, the VPE and MBE are mentioned as the main epitaxy technique. Why do these models not focus on liquid phase epitaxy? And the MOVPE is also mentioned many times instead of VPE, should these processes be strictly differentiated?

3. All data is simulated, and the manuscript offers two experiments by other researchers. More experimental data with good agreement are needed to prove the validity and reliability of the model. It would be better to provide statistical tables or graphs.

4. The authors should pay attention to the standardization and alignment of the figures. The growth time is applied as abscissa in figures 4, 7, and 8, which should be unified in writing. Other coordinate titles, such as “Nanowires length”, also have the same problem.

5. The authors should check the spelling and formatting in the manuscript carefully. The words of “untapered NWs” are mistakenly written as “untapered MWs” and “unrapered NWs”. And the roman numerals are sometimes written in formula format, while others are in text format.

6. The growth condition is very important for Nanowires. The researchers have made great progress in the growth. Additionally, doping also affects the growth condition. The authors should cite the references (Science Advances, 2021,7, 16: eabf7358; Advanced Materials, 2021, 33, 2104942.)

Author Response

The response letter is attached.

Reviewer 3 Report

This study offers an extensive overview of growth mechanisms and modeling techniques for III-V and III-nitride nanowires (NWs) and NW heterostructures, which hold great potential as building blocks for both fundamental research and applications in nanoelectronics and nanophotonics. However, there are several aspects of the text that require improvement before it can be published:

1) The text lacks sufficient context and background information about the study's purpose, methodology, and significance of the results, which makes it difficult to grasp the relevance of the presented information. The text mentions various crucial aspects of catalyst-free NW growth that warrant further investigation but fails to outline any specific plans for future research or describe how these questions will be tackled. To address this, the study's context should be expanded upon in both the introduction and conclusion sections. Additionally, I recommend discussing the applied theoretical methods' ideological aspects in a more accessible manner, as this will help broaden the potential readership of the publication. In its current state, the paper is highly specialized and geared toward theorists.

2) The work does not emphasize the novelty of its findings, nor does it clarify the distinctions between this study and the author's previous works or similar research conducted by other groups using comparable methodologies. To remedy this, the manuscript should include a discussion of these topics in the introduction.

Author Response

The response letter is attached.

Round 2

Reviewer 3 Report

The work has been significantly improved and can be published in its submitted form.